# The Apparent Activation Energy of a Novel Low-Calcium Silicate Hydraulic Binder

**DOI:** 10.3390/ma14185347

**Published:** 2021-09-16

**Authors:** Mónica Antunes, Rodrigo Lino Santos, João Pereira, Ricardo Bayão Horta, Patrizia Paradiso, Rogério Colaço

**Affiliations:** 1Faculty Instituto Superior Técnico, University of Lisbon, Av. Rovisco Pais, 1049-001 Lisboa, Portugal; monica.h.antunes@tecnico.ulisboa.pt (M.A.); teresa.empis@cires.pt (R.B.H.); patrizia.paradiso@tecnico.ulisboa.pt (P.P.); rogerio.colaco@tecnico.ulisboa.pt (R.C.); 2CIMPOR—Cimentos de Portugal, SGPS S.A., Praceta Teófilo Araújo Rato, 2600-540 Alhandra, Portugal; jpereira@cimpor.com; 3IDMEC, Instituto Superior Técnico, Universidade de Lisboa, Av. Rovisco Pais, 1049-001 Lisboa, Portugal

**Keywords:** hydration, kinetics, activation energy, low-calcium binder

## Abstract

In this work, the apparent activation energy (*E_a_*) of a novel low-calcium binder was, for the first time, experimentally determined, using a calorimetric approach. Additionally, a correlation between the *E_a_*, measured at the acceleration period with the C/S ratio of the hydration product is proposed. The *E_a_* of the prepared pastes was determined through isothermal calorimetry tests by calculating the specific rate of reaction at different temperatures, using two different approaches. When comparing the *E_a_*, at the acceleration period of this novel binder with the one published for alite and belite, we observed that its value is higher, which may be a result of a different hydration product formed with a distinct C/S ratio. Finally, to study the temperature effect on the compressive strength at early ages, a set of experiments with mortars was performed. The results showed that the longer the curing time at 35 °C, the higher the compressive strength after 2 days of hydration, which suggests a higher initial development of hydration products. This study also indicated that the novel binder has a higher sensitivity to temperature when compared with ordinary Portland cement (OPC).

## 1. Introduction

Each year, more than 4 billion tons of ordinary Portland cement (OPC) are manufactured and, by the end of 2050, its production is foreseen to increase to over 5 billion tons a year [1]. However, the production of OPC has a strong negative environmental impact since, for every ton of OPC produced, approximately 0.8 tons of CO_2_ is released into the atmosphere, making this industry responsible for 5–8% of total manmade greenhouse gases [2,3]. Hence, within the scope of the 2030 ONU agenda, the development of hydraulic binders that match the technical, economic, and workability qualities of OPC but allow the reduction of the ecologic footprint represents a target and a challenge to both researchers and the cement industry. At present, a number of scholars in the world are engaged in finding solutions in this research area, as summarized in the works of Scrivener et al. and Gartner et al. [4,5], while others have assessed the sustainability of some proposed solutions, e.g., in recent works of Amran et al. or Bilek et al. [6,7].

In recent years, the CIMPOR group, in partnership with Instituto Superior Técnico, has presented a method for producing a new type of hydraulically active calcium silicate binder, which contains 33% less CaO in the raw meal than the typical Portland clinker formulations, allowing a reduction by more than 25% of the process-related CO_2_ emissions [8,9,10,11,12]. The production concept of this new material allows a reduction of calcium incorporation in the mix and opens the possibility for the electrification of the process, which, in turn, leads to further possible significant savings in terms of greenhouse gas emissions.

The novel binder consists of an amorphous calcium silicate material with a C/S molar ratio of 1.1 [12] and, compared with traditional OPC, presented the following characteristics when alkali activated with a waterglass solution: a competitive mechanical performance, lower heat of hydration, and higher workability with lower water/solid ratios [11].

In order to better understand the properties and the hydration mechanisms of this novel binder, an important parameter to study is the activation energy of the pastes either hydrated with water or with the activating solution.

When a hydraulic binder comes in contact with water or an activating solution, it undergoes a series of reactions that causes heat release and the hardening of the material. The study of this rise in temperature can give important information about OPC pastes and concrete, as well as alkali-activated materials. For example, it can help to understand the sensitivity of concrete hydration processes in relation to temperature and allows the estimation of compressive strength of cement pastes at an early age [13]. Additionally, its study on alkali-activated materials such as slag allows the prediction of how high-temperature curing can affect the acceleration of the reaction [14].

Therefore, to capture the temperature sensitivity of a particular mixture, the Arrhenius equation (Equation (1)) can be used.
(1)$$k=Ae^{\frac{-E_{a}}{RT}}$$
where “*k*” represents the reaction velocity constant, “*A*” the pre-exponential constant, “*E_a_*” the apparent activation energy, “*R*” the perfect gas constant, and “*T*” the temperature.

This equation correlates the kinetics of the mixture (*k*) with the apparent activation energy (*E_a_*), which refers to the minimum amount of energy required for the occurrence of a particular reaction, which, in this case, is the hydration reaction.

The *E_a_* represents an important parameter to characterize the sensitivity of the concrete hydration processes to temperature and allows the prediction of the compressive strength of OPC concrete. If both the *E_a_* and the effect of time in temperature are taken into account, the physical and mechanical properties of OPC can be predicted [15].

Typical *E_a_* values for the aqueous dissolution of oxide and hydroxide minerals range from 19 to 86 kJ/mol [16], whereas studies on the *E_a_* of blast furnace slag with different compositions of alkaline activators indicate that *E_a_* can range from 39.2 kJ/mol [17] to 75.2 ± 6.7 kJ/mol [14].

*E_a_* can be calculated by using the method of the single linear approximation of reaction rate, in which the reaction rate of the mixture is calculated by applying a best-fit line of the linear acceleration period on the isothermal test [18].

Linearizing Equation (1), we obtain Equation (2).
(2)$$\ln\left(k\right)=\ln\left(A\right)-\frac{E_{a}}{R}\frac{1}{T}$$

Huang et al. observed that by mixing 4% of portlandite (CH) with alite (C_3_S), there is an increase in the *E_a_*, from 35 kJ/mol to 36 kJ/mol, which indicates that the CH presents a higher energy barrier for nucleation of hydration products than alite. On the other hand, when mixing C_3_S with calcium silicate hydrates (C-S-H), the *E_a_* is reduced to 34 kJ/mol; this occurs because the C-S-H has an auto-catalytic effect on the nucleation of hydration products; therefore, less energy is required for the precipitation of the specimen [19].

Thomas et al. [20] studied the kinetic mechanisms and activation energy for belite (C_2_S), observing that the activation energy for neat pastes of β-C_2_S was 32 kJ/mol, while for a more reactive form of belite, the activation energy increased to 55 kJ/mol. This difference was related to distinct rate-controlling steps for the hydration process. While the lower value was attributed to the dissolution of C_2_S, the higher value was suggested to be related to the nucleation and growth of the hydration products. In the same study, Thomas et al. [20] studied how the presence of sodium silicate in the hydration solution of a paste made of neat belite influenced the belite hydration mechanism. A two-step behavior was observed, which included an initial stage, before the peak of maximum heat release, where the *E_a_* was approximately 50 kJ/mol, and a second stage after the peak, when the activation energy dropped and stabilized around 32 kJ/mol, thus indicating that the hydration process was controlled initially by nucleation and growth mechanism, and finally, by dissolution.

Mehdizadeh et al. [17] investigated how the presence of sodium affected the *E_a_* of activated phosphorous slag. The tests were performed with activators with different ratios of SiO_2_/Na_2_O and Na_2_O/Al_2_O. They reported that by decreasing the silica content and increasing the alkaline ratio, there is a decrease in the *E**_a_*, from 44.5 kJ/mol to 39.2 kJ/mol. Joseph et al. [14] performed a similar study on the apparent activation energy of ground granulated blast furnace slag and class F fly ash, using NaOH and Na_2_SiO_3_ as activators; they observed that the reaction mechanism is governed by chemically controlled reactions and reported an *E**_a_* that ranged from 48.2 ± 5.9 kJ/mol to 75.2 ± 6.7 kJ/mol, depending on the method used to calculate it. Previous studies [21] with blast furnace slags activated with a solution with an alkali modulus (SiO_2_/Na_2_O ratio) of 1.5 and a Na_2_O concentration of 4%wt also reported similar values.

The aim of this work is to calculate the *E_a_* of a novel low-calcium hydraulic binder, to better understand its hydration reaction and temperature dependence. Additionally, to consolidate these results, an experimental study on the temperature effect in the compressive strength at early ages of mortars was performed. The *E_a_* calculations were achieved by using the Arrhenius equation, implementing two different approaches.

Therefore, for the first time, the activation energy of the novel low-calcium hydraulic binder was calculated. The binder was hydrated with either water or an activating sodium silicate solution at different temperatures, ranging from 20 to 35 °C. The results were compared with previous works in order to be able to relate the CaO/SiO_2_ ratio of the hydrated product with the *E**_a_* of the reaction.

## 2. Materials and Methods

### 2.1. Preparation of the Binder

Common Portland cement clinker raw materials were used in the production of the novel amorphous calcium silicate hydraulic binder, with the overall chemical composition being adjusted to a total C/S molar ratio of 1.1. The raw materials were ground, mixed, and compressed in a disc. The compressed disc was broken into four pieces and placed in a silicon carbide crucible. The filled silicon carbide crucible was placed in an electric furnace and heated at a rate of 25 °C/min, to 900 °C, maintaining this temperature for 1 h, to allow the saturation of the chamber atmosphere with CO_2_. Then, the furnace temperature was raised to 1550 °C and kept for 1 h to guarantee the complete melting of the powder mix and its chemical homogenization. Finally, the amorphous material was quenched by pouring the molten mixture into a water container. The amorphous hydraulic binder produced was then ground with a ring mill for 3 min, obtaining a final Blaine fineness around 6000 cm^2^/g.

### 2.2. Preparation of the Sodium Silicate Solution

The sodium silicate solution was obtained by equilibrating a solution of Na_2_SiO_3_ (Na_2_O: 7.5–8.5% SiO_2_, 25.5–28.5%, Chem Lab, Zedelgem, Belgium) with NaOH (98.2%, VWR-Prolabo, Matsonford, PA, USA) to achieve a SiO_2_/Na_2_O modulus of 1.2. The overall content of Na_2_O in solution, with respect to the binder, was defined as 2%wt, and extra deionized water was added in order to achieve a water/solids ratio of 0.35.

### 2.3. Calorimetric Analysis

The isothermal calorimetry tests were performed on a TAM Air instrument (Waters Sverige AB, Sollentuna, Sweden). Pastes prepared with the amorphous binder, and a water/solids ratio of 0.35 were inserted into vials ex situ; therefore, the first exothermic peak characteristic of the wetting of the amorphous binder is not displayed, and the correct heat released was only considered after the first 45 min to allow for the equipment to stabilize. The experiments were performed at 20, 25, 30, and 35 °C. These temperatures were chosen accordingly to the characteristics of the equipment and the qualities of the binder.

### 2.4. Preparation of the Mortars

Five different mortars were prepared with this new binder. Four of them were cured at 35 °C for 2 h, 6 h, 22 h, and 44 h under relative humidity (HR) conditions above HR90% and then transferred to another cabinet at 20 °C and HR95% until they were mechanically tested. The other mortar was cured at 20 °C and HR95% during the entire duration of the experiment. All mortars were alkali activated with sodium silicate, using a solid/water ratio of 0.365; SiO_2_/Na_2_O modulus of 0.9; Na molarity of 3.5 M and overall content of Na_2_O in solution, with respect to the binder, of 3.2%wt.

After 48 h, compressive strength tests were performed on all mortars. Furthermore, mortars cured at 20 °C and those cured at 35 °C for the first 44 h were also tested at 7 and 28 days of hydration.

## 3. Results

The heat development rate of the pastes was studied at 20, 25, 30, and 35 °C in order to correctly employ Equation (2). For pastes hydrated with either water or an alkaline activator, the calorimetric curves and the accumulated heat in the function of time are presented in Figure 1.

A best-fit line was adjusted to the acceleration period in order to obtain the “*k*” term of Equations (1) and (2) through the slope of the fitting equation. The fits were performed in all the experiments made either with pastes hydrated with water or with alkali-activated pastes. Figure 2 presents an example of a calorimetric curve with the respective best-fit linear analysis for the 35 °C alkali-activated sample.

Figure 3 correlates ln(*k*) with 1/*T* for each temperature and hydration condition. The fit for alkali-activated pastes at different temperatures is depicted as a red dashed line, while the fit for pastes prepared with water is shown as a blue dashed line. The activation energy values were determined by using the expressions 3 and 4, which represent the fitting equations obtained from Figure 3, and the values of 82 kJ/mol and 85 kJ/mol were obtained for the alkali-activated pastes and for the pastes prepared with water, respectively.
Alkali-activated (red line): ln(*k*) = −9818.9 (1/*T*) + 16.068   *E_a_* = 82 kJ/mol(3)
Water (blue line) ln(*k*) = −10255 (1/*T*) + 14.969                    *E_a_* = 85 kJ/mol(4)

Another method that can be applied to calculate the *E_a_*, previously used by Joseph et al. [14], is the rearrangement of the Arrhenius equation (Equation (1)) to obtain the instantaneous activation energy. This is carried out by writing the Arrhenius equation at two different temperatures, *T*_1_ and *T*_2_, and then subtracting them.

Then, to obtain the same behavior of the system at different curing temperatures, the concept of equivalent age is used. This represents the time at which both systems have the same degree of reaction (α) [22]. The α can be calculated using the expression (5), where *Q*(*t*) is the cumulative heat released, and $Q_{\infty }$ is the heat release at an infinite time.
(5)$$\alpha =\frac{Q\left(t\right)}{Q_{\infty }}\;\left(5\right)$$

Therefore, if we define $\tau \left(T_{2}\right)$ and $\tau \left(T_{1}\right)$ as the equivalent age at which the degree of reaction is identical in both systems, we obtain expression (6) as follows:(6)$$\tau \left(T_{2}\right)=\tau \left(T_{1}\right)\cdot \exp\{-\frac{E_{a}}{R}\cdot \left(\frac{1}{T_{2}}-\frac{1}{T_{1}}\right)\}$$

Hence, to be able to apply Equation (6), the equivalent age in the activated binder systems of 25 °C and 35 °C, was calculated using Equation (5), at different times of the acceleration period. Since the remaining parameters of Equation (6) are known constants (*R*, *T*_1_, and *T*_2_), it is possible to relate the *E_a_* to α. The results are presented in Figure 4.

To experimentally observe the impact that the temperature has on the compressive strength of the binder, five mortars were alkali activated with sodium silicate and cured for different durations at 35 °C, followed by a curing treatment at 20 °C until being tested. The results obtained after 2 days of hydration are presented in Figure 5.

Compressive strength tests at 7 and 28 days of hydration were also performed on two mortars, initially cured for 0 h and 44 h at 35 °C, and then at 20 °C. The results are presented in Figure 6.

## 4. Discussion

It is broadly accepted by the scientific community that cement hydration is a reaction that undergoes a dissolution–precipitation process [23]. In the first stage of the hydration, the slow observed kinetic is related to the ionic dissolution, which represents the initial rate-controlling step. During this period, the ionic concentration in solution increases until it reaches a saturation point and an equilibrium phase, less soluble than the precursor phase, starts to precipitate, which, in this case, are the C-S-H products. This stage occurs at the so-called acceleration period, measured by isothermal calorimetry, in which the controlling step is the heterogeneous nucleation and growth of C-S-H [19,24]. This type of exothermic reaction is responsible for a large amount of heat release, which can be deleterious on large volume concrete structures, such as dams, since the self-generated heat accumulates within the structure, causing fractures due to localized thermal expansion and decreasing its lifespan [25].

Traditional OPC, after 72 h of hydration, releases more than 250 J/g of cumulative heat [26]. In contrast, in this work, we observed that the hydration of the novel binder, for the same period of time, releases a cumulative heat that is only one-sixth of this value, when hydrated with water, and one-third when activated with the alkaline solution, as seen in Figure 1; however, the strength developed by this binder is still competitive with OPC from the 7th day of hydration and further [11].

Two different approaches were used in the present work to calculate the *E_a_* of the nucleation and growth mechanism at the acceleration period of the reaction. In the first approach, based on Equation (2), which relates the rate of the reaction with the temperature, values of 82 kJ/mol and 85 kJ/mol were obtained for the activated and non-activated pastes, respectively. This further indicates that the measured system is far from a dissolution-controlled regime since it is known that the dissolution of a given material is highly dependent on the pH. This is supported by the work of Li et al., whose results indicate that the pH has a significant effect on the value of the *E_a_* for systems of alkali-activated slag in the range between pH 12 and pH 13.6, revealing an *E_a_* at 1 day in the range of 70–96 kJ/mol for a pH 12 and around 3 kJ/mol for a pH 13.6 [27].

For the second method, based on Equation (6), which correlates the *E_a_* with the degree of the reaction, a range of values from 63 to 80 kJ/mol were obtained in the region of the acceleration period. The small discrepancy in the calculated *E_a_*, when compared to the first approach, has been already reported by Joseph et al. [14], who correlate *E_a_* with the use of different methods of calculation. Furthermore, we can observe that the range of *E_a_* calculated stands above the dissolution *E_a_* of wollastonite (54.7 kJ/mol [16]), which further indicates that this value is not correlated with a dissolution mechanism. This assumption results from the knowledge that wollastonite is a non-hydraulic phase [23], contrary to the present amorphous binder, which dissolves to further precipitate C-S-H products [10,11]; hence, its *E_a_* for dissolution should be lower than that presented for wollastonite.

Therefore, the obtained *E_a_* values can be translated as the energy necessary to provide to the system so that the nucleation and growth of C-S-H can occur [19].

When comparing the *E_a_*, at the acceleration period of this novel binder with alite and belite, 51 [28] and 55 kJ/mol [20], we can observe that the values of 82–85 kJ/mol are considerably higher. This may be due to the formation of structurally different hydration products. In fact, previous works have reported that this material, instead of producing a C-S-H with a C/S ratio of 1.7 [29], such as alite and belite, produces a C-S-H with a ratio of 1.1 [30]. It is known that the mean silicate chain length of a C-S-H structure increases as the Ca/Si ratio decreases. Hence, we hypothesized that the higher the polymerization degree required to form those C-S-H structures, the higher the energy input necessary, translating into higher values of *E_a_*. Additionally, because this binder has the particularity of producing tobermorite 9 Å, [30], more energy will be required in its growth, since this material has highly polymerized silica chains.

The addition of sodium silicate, which causes an increase in alkalinity, showed a considerable impact on the first hours of reaction, decreasing the slow reaction period, characteristic of dissolution. This is a consequence of the influence of pH since its increase causes a rise in the solubility of [SiO_4_]^4−^ ions [31], but also due to the addition of more silica units into the system, allowing the dissolved calcium ions that are in majority to rapidly react and produce C-S-H.

The use of this salt also decreases the *E_a_* from 85 to 82 kJ/mol. This small decrease may be a result of the presence of [SiO_4_]^4−^ that facilitates the nucleation and growth of the C-S-H, similar to what was reported by Thomas et al. [20], which showed that the *E_a_* of belite decreased from 55 kJ/mol to 50 kJ/mol when preparing pastes with water or with a solution of sodium silicate, respectively.

The proximity of the *E_a_* values (for the nucleation and growth rate-controlling step) in the studied systems may indicate that a similar C-S-H is formed by using either water or the alkaline solution. In fact, this theory can be sustained by the work of Santos et al. [11], which reported the main hydration product for both pastes resulted in a C-S-H type phase with a semi-crystalline nature, with the crystalline part of this hydration product resembling an ordered tobermorite-like structure.

Finally, to discuss the temperature effect on the early age compressive strength of the binder, first, we need to compare it with the experimental results obtained by Ezziane et al. [32]. In this study, mortars made with OPC were cured under constant temperatures of 20, 40, and 60 °C, and no significant changes were found by the authors. If we contrast these results with the mortar’s compressive strength obtained with the novel binder (Figure 5 and Figure 6), and we confirm that the hydraulic reaction of the novel binder has a higher sensitivity to temperature. This was expected since the binder has a higher *E_a_*. In fact, the longer the curing times at 35 °C, the higher the compressive strength results obtained. This increase in early age mechanical performance suggests a higher development of hydration products, which may be a result of the increase in solubility enhanced by the higher temperatures. Since the dissolution process is the initial rate-controlling step, by facilitating this process, the ionic concentration in the solution that is necessary to reach saturation point and start precipitation is anticipated, which translates into higher initial strengths.

Therefore, the higher temperature sensitivity that this novel binder presents can be used to adjust and optimize the reaction mechanism and kinetics of the hydration reaction to a given situation, such as increasing the rate of cement hydration, decreasing setting time, or even accelerating its compressive strength performance. With this purpose, further temperature sensitivity tests must be performed to be able to optimize the amount of time that the hydration should occur at higher temperatures and correlate it to the physical properties shown on the hydrated binder.

## 5. Conclusions

In this work, the activation energy of a novel low-calcium hydraulic binder was experimentally obtained using calorimetric experiments at 20, 25, 30, and 35 °C. The obtained results revealed the following conclusions:The *E_a_* correlates with the acceleration period and can be translated as the energy necessary to provide to the system so that the nucleation and growth of C-S-H can occur;The experimental *E_a_*, calculated for this new amorphous hydraulic binder (82–85 kJ/mol) is higher than that of alite or belite (51 and 55 kJ/mol). This difference in *E_a_* may result from the formation of structurally different hydration products, with a lower C/S ratio of 1.1;It was hypothesized that a higher *E_a_* at the acceleration period may translate into a higher mean silicate chains length of C-S-H with a lower C/S ratio;The similarity of the activation energy value determined for the alkali-activated the non-activated pastes may indicate the formation of similar C-S-H products. The small decrease may be a result of the presence of [SiO_4_]^4−^ species, which facilitates the nucleation and growth of the C-S-H;The compressive strength tests on mortars cured for different periods of time at a higher temperature suggest that this novel binder has a higher sensitivity to temperature when compared with OPC;The increase in temperature promotes a faster dissolution of the binder, which allows for the formation of a more hydrated product at early ages;A low amount of released heat after 3 days of hydration was reported in this novel binder, i.e., (45 J/g, when hydrated with water, and 100 J/g when hydrated with the activating solution). Previous studies have confirmed that, even with this low amount of heat release, the binder is able to obtain competitive strengths, which opens the possibility of producing a hydraulic binder with a compressive strength comparable to that of OPC but with a lower CO_2_ footprint and improved hydration heat properties.

## Figures and Tables

**Figure 1 materials-14-05347-f001:**
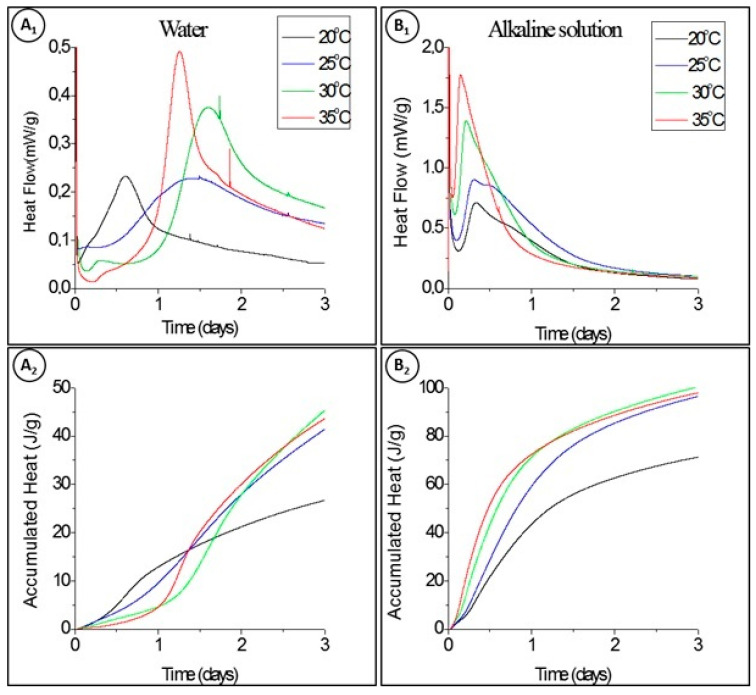
Heat release (**A_1_**,**B_1_**) and accumulated heat release (**A_2_**,**B_2_**) in function of time at different temperatures, with different hydration solutions: water (**A_1_**,**A_2_**) and alkaline solution (**B_1_**,**B_2_**).

**Figure 2 materials-14-05347-f002:**
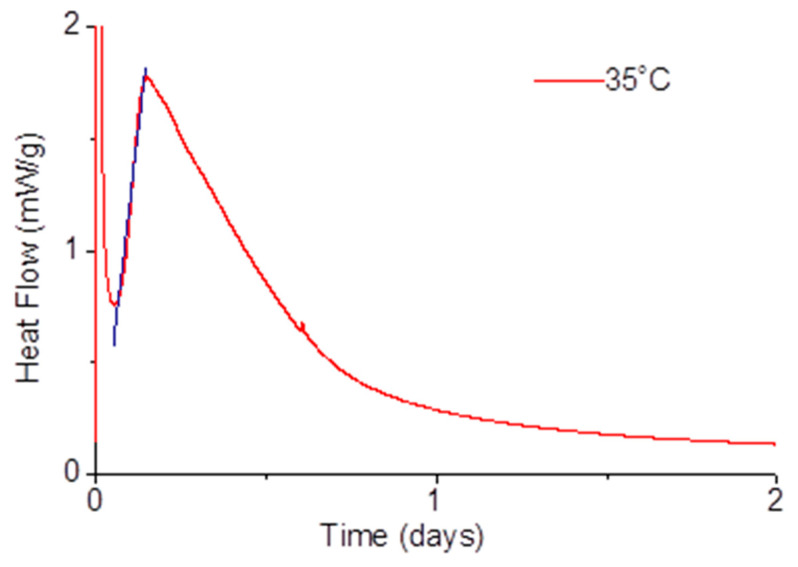
Heat release of a paste activated with Na_2_SiO_3_ and a water/solid ratio of 0.35 at 35 °C, with a best-fit line on the linear acceleration period (blue line).

**Figure 3 materials-14-05347-f003:**
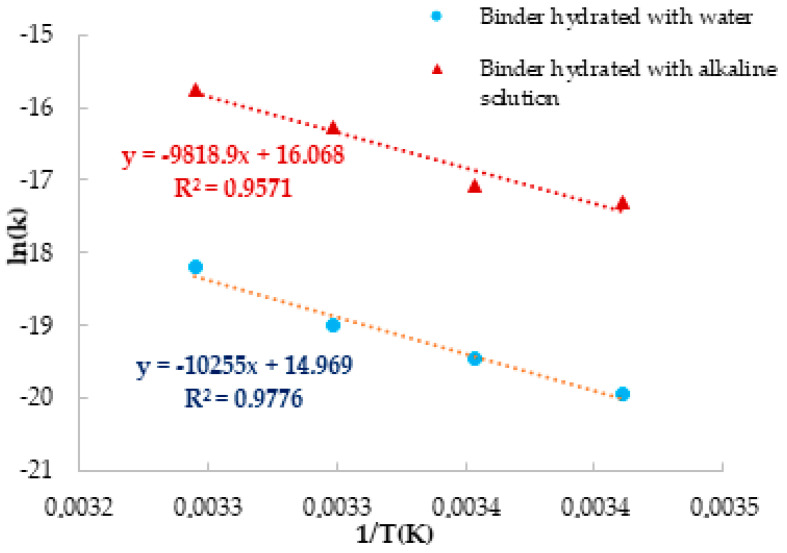
Graphic representation of ln(*k*) in function of 1/*T* for the temperatures 20, 25, 30, and 35 °C. The blue dots correspond to the pastes hydrated with water, while the red triangle corresponds to the alkali-activated pastes. The best fit is showed as dashed lines.

**Figure 4 materials-14-05347-f004:**
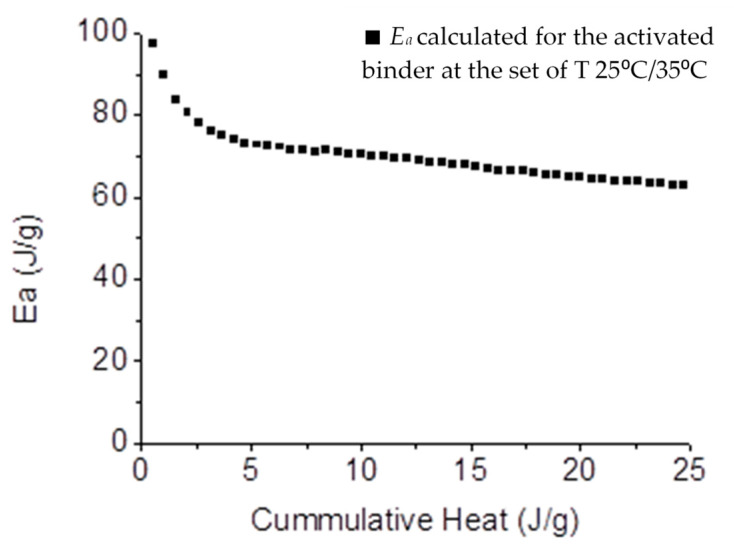
Evolution of the *E_a_* of the activated binder determined by Equation (6), at the set of temperatures 25 °C and 35 °C.

**Figure 5 materials-14-05347-f005:**
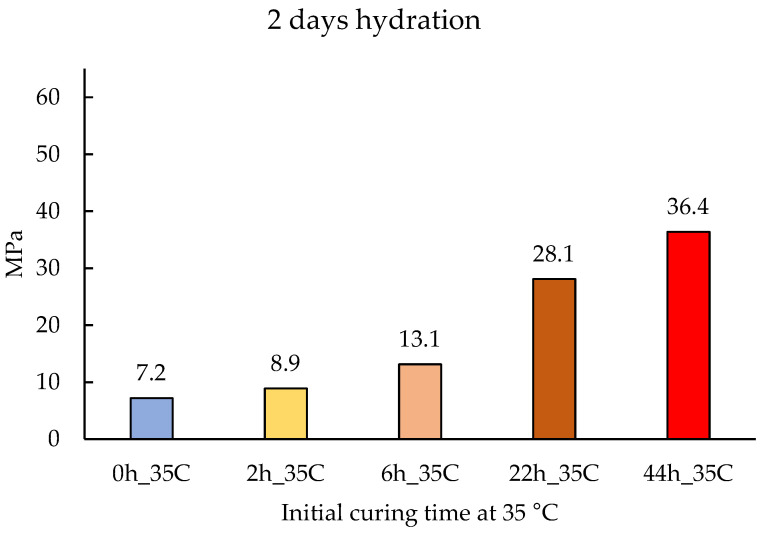
Compressive strength of alkali-activated mortars at 2 days of hydration. The curing times at 35 °C were 0 h, 2 h, 6 h, 22 h, and 44 h, followed by curing at 20 °C to complete the 2 days.

**Figure 6 materials-14-05347-f006:**
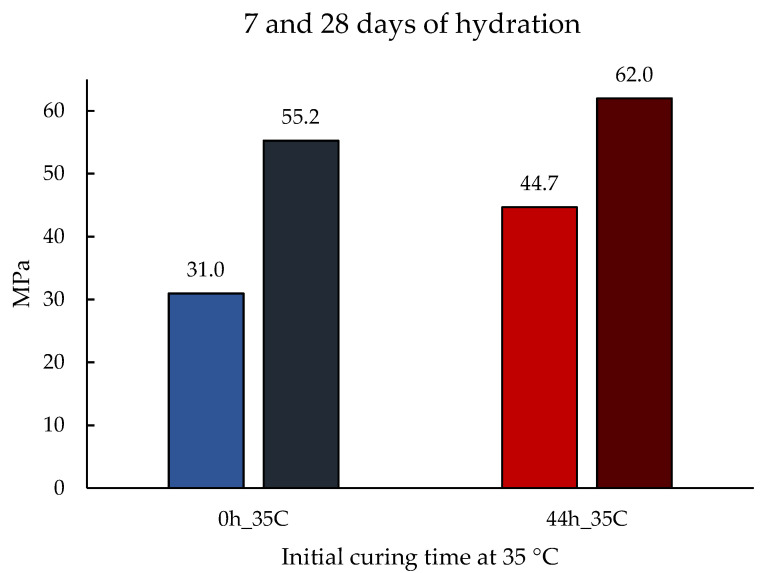
Compressive strength of alkali-activated mortars at 7 (light blue and light red) and 28 days (dark blue and dark red) of hydration. The curing times at 35 °C were 0 h and 44 h, followed by curing at 20 °C to complete the 7 and 28 days.

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
