# Peer review of "The Apparent Activation Energy of a Novel Low-Calcium Silicate Hydraulic Binder"

_materials, 2021, doi:10.3390/ma14185347_

Round 1

Reviewer 1 Report

The   development   of   hydraulic   binders   which   match   the   technical,   economic   and workability qualities of OPC, but allow the reduction of the ecologic footprint, represents a target and a challenge toboth researchers and the cement industry.The novelty of this work is the determination of activation energy, based on the Arrhenius  equation,  for  a  novel  low  calcium  hydraulic  binder. In this work the activation energy of a novel low-calcium hydraulic binder was experimentally obtained usingcalorimetric experiments at 20, 25, 30 and 35⁰ C. The obtained results  showed  that  the calculated Eacorrelates with  the acceleration  phase. It wasalsoobserved that the experimental Ea, calculated for this new amorphous hydraulic binder (82-85 kJ/mol),is higher than that of alite or belite 307(51 and 55 kJ/mol). This difference in Eamay be a result of the different hydration products formed with a lower C/S ratio of 1.1.     Higher Eaat  the  acceleration  phasemay  be  relatedto highermean  silicate  chains length of C-S-H with alower C/S ratio.

Very interesting work. Of interest to readers 

Reviewer 2 Report

Suggestions are contained in the attached PDF file.

Reviewer 3 Report

The work and topic are current.
The manuscript can be potentially interesting for readers of the Materials journal.
The manuscript has the usual structure.

At present, a number of authors in the world are engaged in the solved research area. In particular, the ability to reduce the impact and production of CO2 is very important for sustainable development.
The addressed area includes a number of aspects and approaches, where recent articles include:

Bilek, V.; et. al. Frost Resistance of Alkali-Activated Concrete—An Important Pillar of Their Sustainability. Sustainability 2021, 13, 473. 
Amran, M.; et.al.  Fly Ash-Based Eco-Efficient Concretes: A Comprehensive Review of the Short-Term Properties. Materials 2021, 14, 4264.

I recommend rewriting and expanding the introduction section.

In the discussion and conclusion, the authors should focus more on formulating new knowledge and benefits for further research.
The manuscript must be revised before publication.
Authors must also carefully prepare and review the manuscript according to the MDPI template. 
For example, all references are in the wrong format.

Reviewer 4 Report

Please find attached a PDF file with my comments and suggestions for authors.

Round 2

Reviewer 2 Report

Judging by the short time the editors have given me to respond to the changes made by the authors, my review is intended to be a mere formality and publication is a foregone conclusion. Since the weaknesses I have pointed out in the article, only some of which have been superficially corrected, do not bother the editors, and even less do the ethical issues I have pointed out, let formalities be satisfied. If he believed that it made any difference, I would again tick the 'reject' option....

Reviewer 3 Report

The authors reworked the manuscript. 
The revised manuscript has a significantly greater informative value.
All important information in the manuscript is given. 
The visual of the manuscript has also improved significantly.
Information on the methods and results used is extended. 

The solved area of activation energy of a novel low-calcium silicate hydraulic binder is paid attention, when the manuscript sufficiently and interestingly expands the knowledge in the field of physical-mechanical properties. 
The manuscript is suitable for publication and will be potentially interesting for readers of Material journal.